# Mastication inefficiency due to diminished or lack of occlusal support is associated with increased blood glucose levels in patients with type 2 diabetes

Yeter E. Bayram[1], Mehmet A. Eskan[2]*

1 Department of Internal Medicine, Hamidiye Sisli Etfal Education and Research Hospital, Istanbul, Turkey,
2 Department of Periodontics and Endodontics, University at Buffalo School of Dental Medicine, Buffalo, NY, United States of America

* mehmetes@buffalo.edu

## Abstract

**Data Availability Statement:** All relevant data are within the paper and its Supporting Information files.

### Background

It has been shown that mastication may contribute to a lower risk of diabetes, and occlusal support reduced the risk of diabetes by improving glucose metabolism after meals. However, the relationship between inefficient mastication and blood glucose levels in patients with type 2 diabetes (T2D) remains unclear. This retrospective study, therefore, aimed to investigate the association between mastication inefficiency due to diminished occlusal support and blood glucose control in subjects with T2D.

### Methods

Ninety-four subjects (mean of 54.9 years) were recruited in this study. Subjects with at least 1-year T2D medical history and current medications for T2D were included. Subjects were divided into 2 groups: The control group (41 subjects) included Eichner group A (4 occlusal functional areas in the posterior area). The test group (53 subjects) included Eichner group B (1–3 occlusal functional areas) and group C (no natural occlusal contact). Blood glucose level was significantly lower in the control group participants than in the test group. Subject (s) showing diminished or lack of occlusal support and requiring a fixed restoration were treated with an implant-supported fixed restoration. These groups' levels of glycated hemoglobin (A1c) were compared using the independent student t-test.

### Results

Blood glucose level was significantly lower in the control group (7.48) as compared to those in the test group (9.42). The mean differences between the two groups were 1.94 ± 0.39 ($p$ = 0.0001). Differences in white blood cell counts and body mass index (BMI) were not statistically significant between groups. Blood glucose levels could be reduced (from A1c 9.1 to 6.2) following a fixed implant-supported restoration in T2D patients with diminished occlusal support.

**Funding:** The author(s) received no specific funding for this work.

**Competing interests:** The authors have declared that no competing interests exist in this study.

## Conclusion

The results suggested that masticatory inefficiency due to diminished dental occlusion was associated with an increase in poor controlled-blood glucose levels among T2D patients.

## Introduction

Diabetes is a chronic and broad-spectrum metabolic disorder characterized by hyperglycemia, which occurs due to relative or absolute insulin deficiency or insulin resistance developed in peripheral tissues. It affects many organs and causes multisystem problems, such as end-stage renal disease and cardiovascular disease [1]. The prevalence of diabetes is increasing world-wide. It is estimated to be 463 million individuals in 2019, rising to 578 million in 2030, and 700 million in 2045 [2, 3]. Type 2 diabetes accounts for 90–95% of all diabetes cases [4]. The therapeutic approach of T2D includes maintaining a healthy weight, healthy diet, regular physical activity, quitting smoking, glycemic control, regulating blood pressure, lowering lipid levels, and the use of specific therapeutic agents are fundamental to treatment in terms of preventing complications among diabetics patients [5]. These approaches are considered cardiovascular and renal benefits and constitute the basic treatment approaches to prevent complications in individuals with diabetes [6]. In addition, continuous education of healthcare professionals and patients is essential to reduce the risk of developing acute or chronic complications of T2D.

Maintaining masticatory function might play a crucial role in general health and reduced or lack of masticatory efficiency is considered a major issue among adults. It has been associated with insufficient nutritional intake, cognitive impairment, increased risk of cardiovascular disease, stroke, all-cause mortality, and obesity [7–9]. Mastication has been shown to regulate blood glucose levels via different mechanisms. People unable to fully masticate due to teeth loss or ill-fitting dentures had insufficient dietary fiber, magnesium, or calcium [10, 11], nutrients that might play a protective role against T2D [12, 13]. Studies have also shown that there is an association between mastication and the prevalence of diabetes with increased T2D in the total edentulous subjects [14, 15].

It has been reported that proper chewing function elicits a lower postprandial plasma glucose concentration by increasing the anorexigenic gut peptide YY and glucagon-like peptide 1 (GLP-1) in the intestinal tissue [16, 17], thereby it can result in increased early-phase insulin secretion following proper chewing function [18]. Activation of the GLP-1 receptor has been shown in reducing hemoglobin A1c, cardiovascular death, and stroke [19]. Accordingly, subjects with a higher masticatory performance showed less occurrence of diabetes [20]. Additionally, nerves in the masseter muscle and periodontal tissues have been reported to be involved in regulating dietary intake [21]. Half of the proprioceptive neurons located in the masseter muscle and periodontal ligament are directly connected to the trigeminal mesencephalic nucleus, without any interruption [21]. These signals are precise and reach the cortex quickly enabling the central nervous system to activate the histamine-1 receptor in the hypothalamus paraventricular nucleus, resulting in less food intake [21].

Mastication, crushing, and grounding bolus, is a complex functional movement involving diverse and accurate mandibular patterns, which is affected by peripheral inputs, including the tongue, teeth, dental occlusion, and muscles, such as the masseter muscle [22]. It was shown that there was a significant correlation between mastication performance and the number of remaining teeth, and the occlusal support [23]. Partial or total edentulism has been shown to

reduce mastication capacity [24]. Even missing only premolar occlusal contact could negatively affect masticatory function [25, 26].

The masseter muscle participates in swallowing, speech, and mastication [27]. Masseter muscle thickness was significantly linked with the mastication performance [28]. Any malocclusion or edentulism can affect the whole mastication system. Several studies have shown that masseter activity could be compromised in subjects with malocclusions, such as the crossbite or open bite malocclusion [29–31]. These studies clearly indicate that inputs from mechanoreceptors through teeth are critical for masticatory muscle health. Integrins, communicators between the cell and extracellular matrix (ECM), are shown to involve in the muscle cell function [32, 33]. Indeed, the production of integrins in the masseter muscle located, on the crossbite side, was found to be reduced significantly as compared to the normal occluding side [32]. This study indicated that the production of integrins, which are involved in the regulation of contractile forces, is regulated by occlusal support.

As aforementioned, there were multiple studies on the association between occlusal support and the prevalence of T2D. However, the association between occlusal support and controlling blood glucose in T2D patients is not adequately addressed in these studies. We, therefore, aimed to determine if mastication inefficiency, due to lack or diminished occlusal support, negatively affects the controlling blood glucose levels among subjects with T2D.

## Methods

Subjects were selected at the endocrinology and/or internal medicine outpatient clinic in a public hospital in Istanbul (Turkey) from March 2020 to July 2022. A total of 94 participants received oral health examinations, blood glucose measurements, and type 2 diabetes assessments in this retrospective study. All subjects between 30 and 75 years old, diagnosed with T2D for more than one year, were included in this study. The levels of A1c were collected from electronic medical records. Subjects who presented at least 1-year T2D medical history and using current medications for T2D were included. Subjects with cognitive disorders preventing communication, a history of type 1 diabetes, heart failure, cancer, liver cirrhosis, chronic kidney disease, rheumatological diseases pregnancy, or cancer were excluded.

Individuals showing total edentulism or missing unilateral and/or bilateral posterior teeth in the mouth were classified based on the Eichner index [34]. Antagonistic occlusal contacts by natural teeth, crowns, or fixed partial dentures were recorded. The subjects were divided into 3 groups: Group A with 4 occlusal functional areas (2 premolars and 2 molars on each side); group B with 1–3 posterior occlusal functional areas; and group C with no functional occlusal contact. The control group corresponded to Eichner's group A (including fixed crown or bridge) and they reported not having any chewing problems. Subjects in the test group included Eichner's group B, or C classification [25, 26] reported that they were unable to chew any solid food properly due to either lack of posterior teeth or an ill-fitting removable (partial or complete) prosthesis. To assess the nutritional and infectious status of the subjects, their body mass index (BMI) and total leucocyte number (WBC) were determined. Among the individuals, who wanted to re-establish dental occlusion were treated accordingly (implant-supported fixed restoration) as mentioned in our previous study [35]. A fixed implant-supported restoration was performed on volunteers. Surgical procedures, including extraction of the teeth and immediate implant placement (Nobel Biocare and Straumann), and delivery of prostheses (immediate fixed provisional/loading and final fixed prosthesis) was performed as described in our previous study [35].

## Statistical tests

The difference in A1c between the test and control groups was the primary outcome. Student's t-test and Fisher's exact test were used in determining these differences. In addition, multiple linear regression analysis was conducted to determine whether BMI affected the baseline of A1c of the subjects. The p values and 95% CIs were estimated in a two-sided manner and $p < 0.05$ was considered significant. Statistical analyses were performed using GraphPad Prism 9.4 software (Boston, MA, USA).

## Ethics statement

This study protocol was approved by the Ethics Committee of Seyrantepe Sisli Etfal Education and Research Hospital (protocol #2125; Date Aug 23, 2022). Written informed consent was obtained from all participants.

## Results

### Demographic data of the subjects

From March 2020 to July 2022, a total of 94 subjects were included in the current study. The characteristic of the subjects is shown in Table 1. Thirty-two (60.38%) and sixteen (39.02%) females were included in the test and control groups, respectively. The number of males was 31 (39.62%) and 25 (60.98%) in the test and the control group, respectively. The mean age of the subjects in the test group (57.15 years) was higher than that in the control group (52.07 years). Regarding lifestyle, the rate of smokers (former and current) was not significant in the test (20.75%) and the control group (31.71%).

We also determined the number of subjects using only oral antidiabetic drug/s (OAD), including metformin only, metformin plus others, or OAD in combination with insulin. The number of individuals using OAD only or OAD with the combination of insulin did not show differences between the groups. (Table 1). The duration that the subjects were diagnosed with T2D was determined. The mean duration of T2D in the test group (11.1 years) was slightly longer as compared to the control group (8.4 years). But the mean differences (2.6 ± 1.5) between the groups were not statistically significant ($p = 0.07$).

It has been well known that BMI is associated with the onset of T2D and might make blood glucose level control harder in diabetic patients [36]. Therefore, we determined whether the subjects' BMI showed any differences at the baseline. We found the mean level of BMIs at the baseline in the test group was very similar to the test group. The mean BMI was 30.19 and 29.61 in the test and control groups, respectively (Fig 1). The mean difference between the groups was 0.57 ± 1.04 ($p = 0.58$). WBCs have been considered in predicting a systemic infection or inflammation status [25]. Systemic infection and/or inflammation have been shown to negatively affect the control of blood glucose levels in T2D patients [24]. The total WBCs in the test and control groups were $7.79 \times 10^9$/L and $7.91 \times 10^9$/L, respectively ($p > 0.05$). The mean difference between the groups was 0.12 ± 0.4 ($p = 0.76$). These results showed that the number of WBCs was not statistically significant between the groups (Fig 2).

It is also known that nephropathy is one of the major complications in patients with T2D [37]. Creatinine levels were 0.78 and 0.76 in the test and control groups, respectively (Fig 3). The mean difference between the groups was 0.02 ± 0.04 ($p = 0.62$) indicating the levels of creatinine in both groups were very similar to each other.

**Table 1. Demographic, lifestyle, medication, and medical history of the subjects.**

|  | Test | Control | P value |
|---|---|---|---|
| **Sex** |  |  | **0.06** |
| Male, n (%) | 21 (39.62) | 25 (60.98) |  |
| Female n (%) | 32 (60.38) | 16 (39.02) |  |
| Age, year, mean ± SD | 57.15 ± 7.92 | 52.07 ± 8.75 | 0.004 |
| Cigarette smoking n (%) | 11 (20.75) | 13 (31.71) | 0.2435 |
| Hypertension, n (%) | 35 (66.34) | 23 (56.10) | 0.3939 |
| Hyperlipidemia n (%) | 23 (43.40) | 12 (29.27) | 0.1989 |
| Ischemic heart disease n (%) | 9 (16.98) | 2 (18.18) | 0.1056 |
| Hypothyroidism n (%) | 4 (66.67) | 2 (33.33) | 0.6933 |
| Medication in use |  |  |  |
| Only insulin, n (%) | 7 (12.96) | 4 (9.76) | 0.7521 |
| Only OAD, n (%) | 27 (50.0) | 27 (65.85) | 0.1461 |
| Insulin + OAD, n (%) | 19 (35.19) | 10 (24.39) | 0.3685 |
| Metformin, n (%) | 14 (25.93) | 11 (26.83) | >0.9999 |
| Metformin + Gliclazide, n (%) | 5 (9.26) | 4 (9.76) | >0.9999 |
| Metformin + DPP4-I, n (%) | 12 (54.55) | 10 (45.45) | 0.8112 |
| Metformin + Pioglitazone, n (%) | 2 (3.70) | 5 (11.63) | 0.2356 |
| Metformin + SGLT2-I, n (%) | 14 (25.93) | 11 (26.83) | >0.9999 |
| Time diagnosed with T2D |  |  |  |
| Mean (year) ± SD | 11.11 ± 7.85 | 8.4 ± 7.01 | 0.0798 |
| 1–5 years, n (%) | 12 (22.64) | 18 (43.90) | 0.0439 |
| 6–10 years, n (%) | 20 (37.74) | 13 (31.71) | 0.6638 |
| > 10 years, n (%) | 16 (30.19) | 8 (19.51) | 0.3404 |

Continuous variables including age and duration of T2D were described in mean and standard deviation (SD). Number and frequencies (%) were used for categorical variables including, cigarette smoking, hyperlipidemia, ischemic heart disease, hypothyroidism, hypertension, oral anti-glycemic drug/s (OAD), and insulin. Student's t-test and Fisher's exact test were used for continuous and categorical variables, respectively. DPP4-I: Dipeptidyl peptidase-4 inhibitor; SGLT2-I: Sodium-glucose co-transporters 2 inhibitors).

### Diminished occlusal support was associated with increased A1c

We next investigated the association between occlusal support and blood glucose levels among T2D subjects. The mean of A1c in the test and control groups was 9.42 and 7.48, respectively. The mean difference between the groups was 1.93 ± 0.39 (95%CI 1.159–2.277, $p = 0001$), indicating blood glucose levels were significantly higher in the test group as compared to the control group (Fig 4). It is well known that the masticatory efficiency has been substantially increased in implant-supported fixed dentures, as compared to conventional removable dentures, in completely edentulous patients [38]. Therefore, a subject who showed a high blood glucose level (A1c = 9.1) and diminished occlusal support (Fig 5) was treated with a fixed implant-supported restoration (Fig 6) as described previously [35]. The patient's A1c was reduced from 9.1 to 7.8 in 4 months, and it reached 6.2 in 18 months of follow-up (Fig 7). Together, these findings suggest occlusal support plays an active role in controlling blood glucose levels in T2D individuals.

## Discussion

This retrospective study showed that inefficient mastication due to diminished occlusal support was associated with poor control of blood glucose levels among T2D subjects. Glycemic

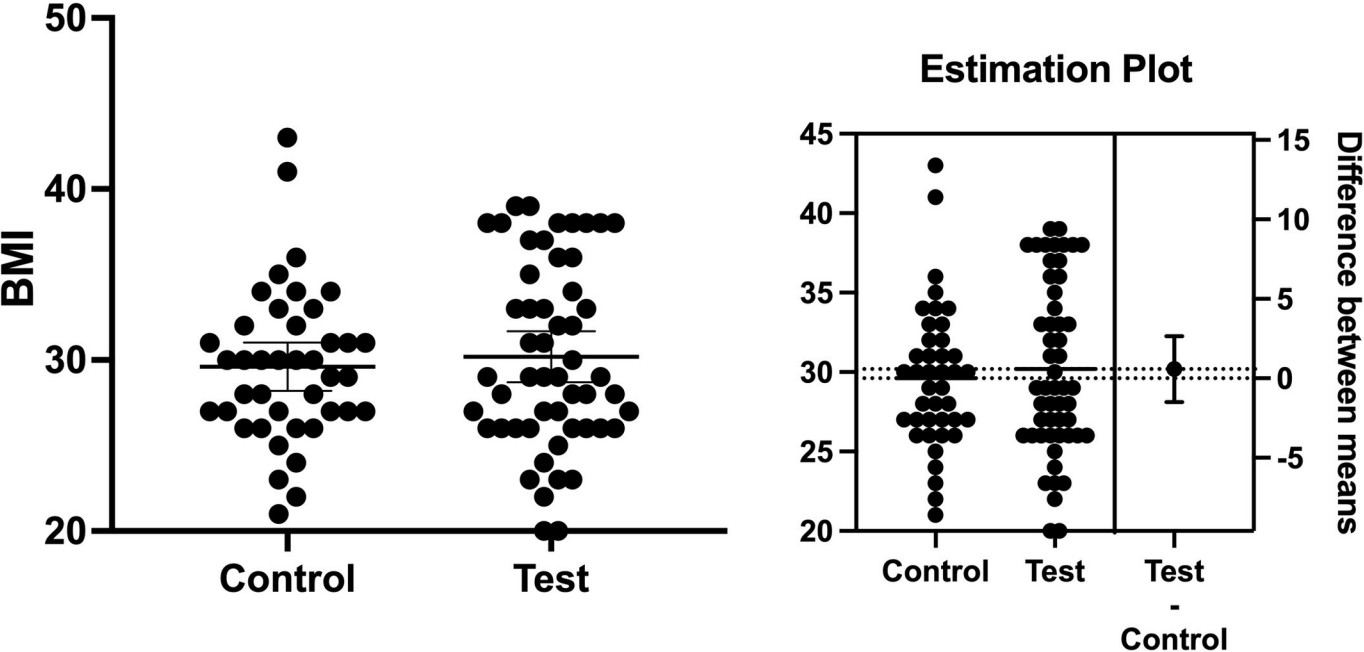

**Fig 1. Body mass index (BMI).** The mean BMI was 30.19 and 29.61 in the test and control groups, respectively. The mean difference between the groups was 0.57 ± 1.04 ($p = 0.58$).

control was poorly maintained in T2D subjects missing posterior occlusal support or using a removable denture. To our knowledge, this is the first study to clarify the association between occlusal support and controlling A1c in patients with T2D.

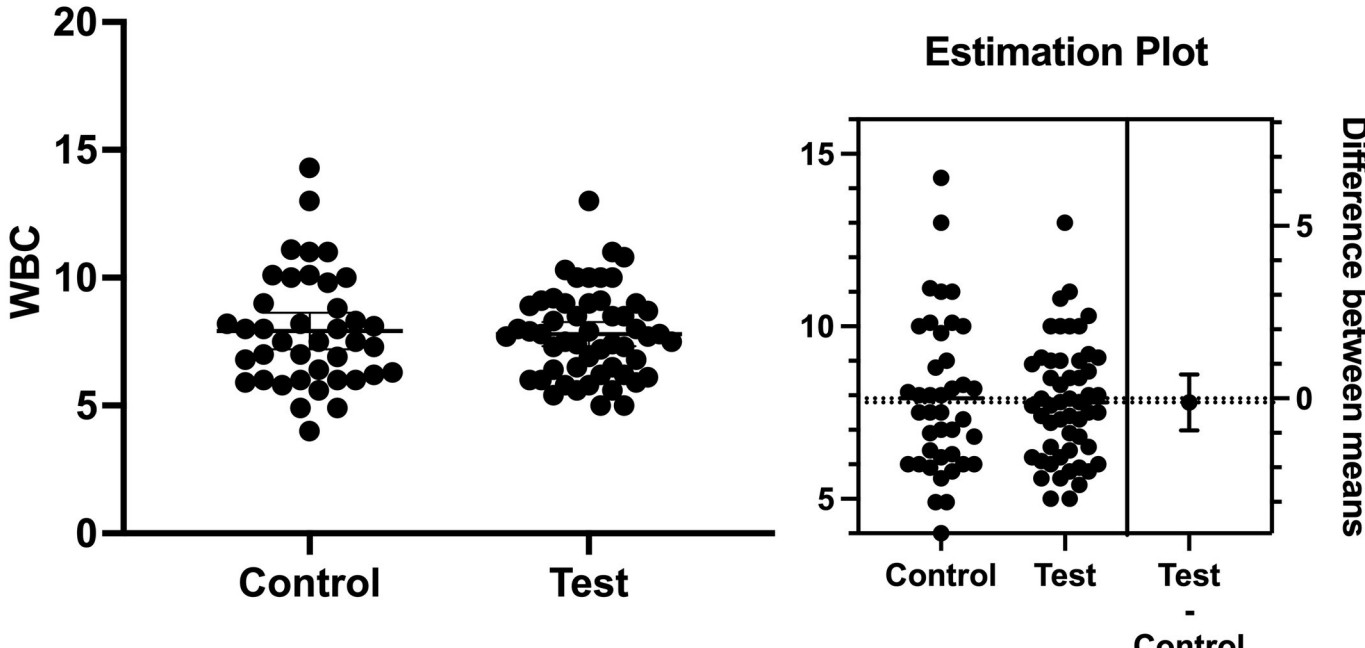

**Fig 2. The number of white blood cells.** The number of white blood cells (WBCs) in the test and control groups was 7.79x10$^9$/L and 7.91 x10$^9$/L, respectively. The mean difference between the groups was 0.12 ± 0.4 ($p = 0.76$).

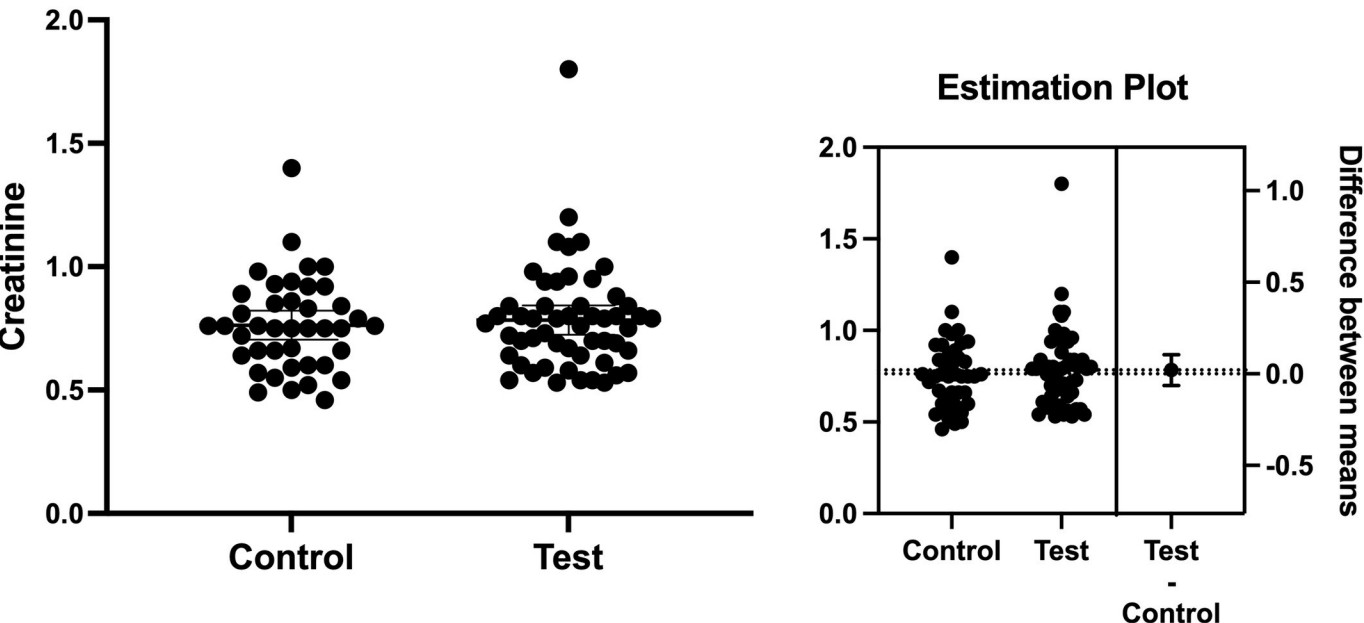

**Fig 3. The level of creatinine in the groups.** Creatinine levels were 0.78 and 0.76 in the test and control groups, with the mean difference between groups (left) being 0.02 ± 0.04 (*p* = 0.62).

Chewing function could be determined by tooth loss, several occlusal support areas, and the Eichner index [25, 39]. There was a close relationship between tooth loss and lack of occlusal support [28, 29]. The maximum bite force was closely related to the functional occlusal area

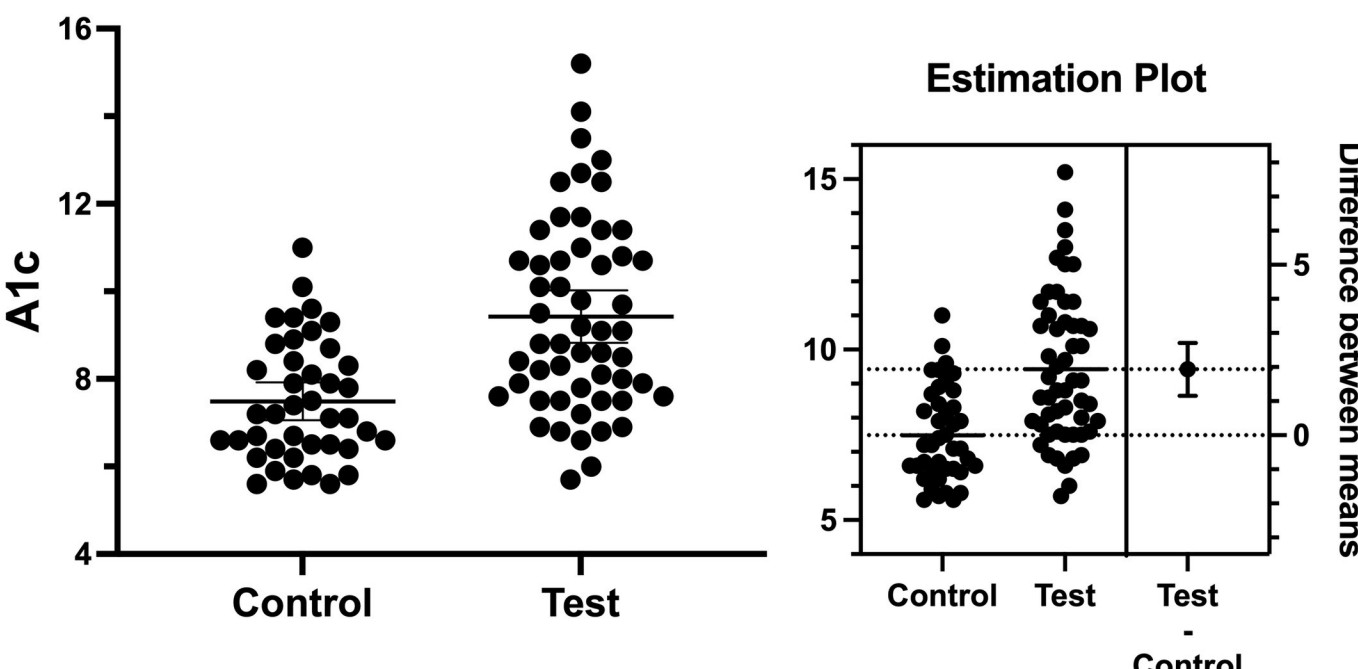

**Fig 4. The value of A1c in both groups.** Mean values of A1c in the test group and control group were 9.42 and 7.48, yielding the mean difference between the groups (left) as 1.93 ± 0.39 (95%CI 1.159–2.277 and *p* = 0001).

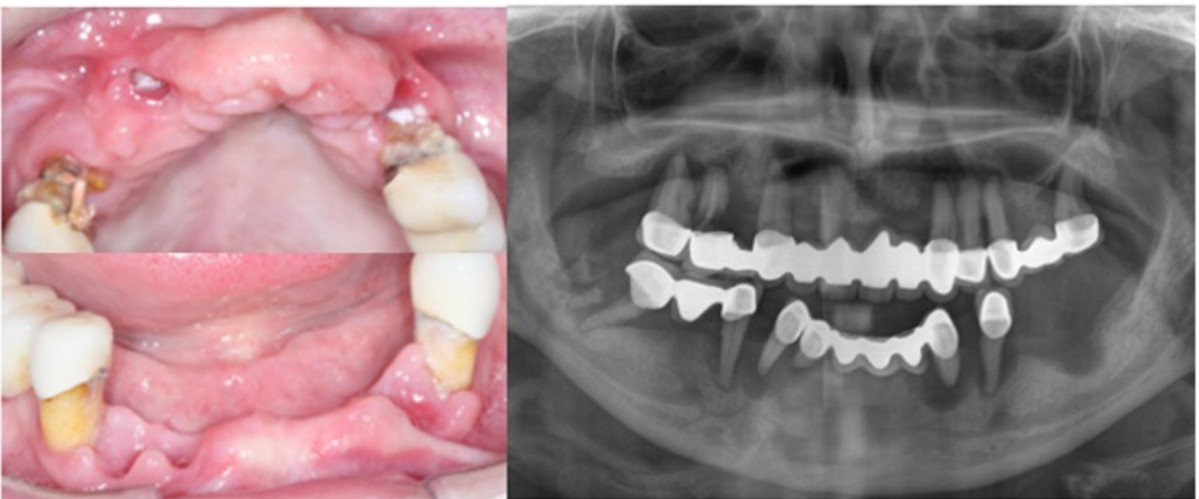

**Fig 5. Pre-operative views.** Initial intra-oral upper and lower jaw (right) and the initial panoramic x-ray (left).

and the Eichner index [40]. People who have significantly impaired mastication efficiency may shift to a soft, reduced fiber, and lower nutritious diet, which could lead to malnutrition in long term. This study showed that the group of patients with diminished posterior occlusal support (Eichner index B or C) or using a removable partial or complete prosthesis tended to have a higher mean A1c level than the control group (Eichner index A). It is known that an increase of 1% in A1c concentration was associated with about a 40% increase in cardiovascular or ischemic heart disease mortality among diabetics patients [41]. A1c values in the test and control groups were 9.44 and 7.48, respectively, resulting in 1.92 ± 0.39 mean differences between the groups ($p = 0.0001$). As aforementioned, the mean difference of 1.92% would significantly affect the general health of diabetic individuals.

About 40–50% of the patients reported a preference for eating liquid or pureed foods, indicating that they are somewhat concerned, as difficulty or dissatisfaction with mastication can lead to dietary restrictions and, consequently, interfere with glycemic control, harming the quality of life of subjects. Reduced chewing function and increased food intake have been

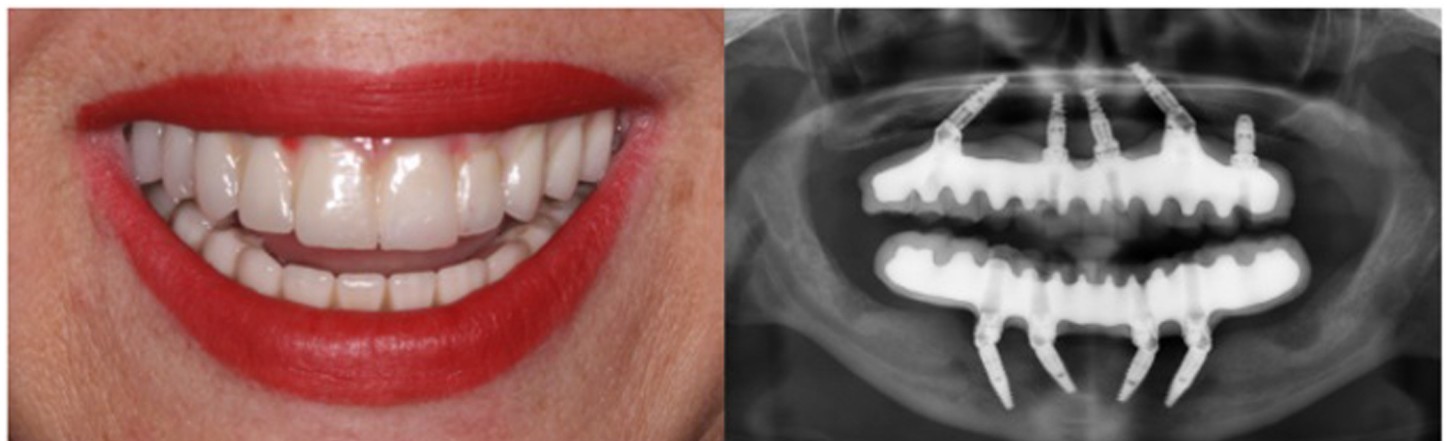

**Fig 6. Post-operative views.** Final view of the subject (right), and panoramic x-ray (left).

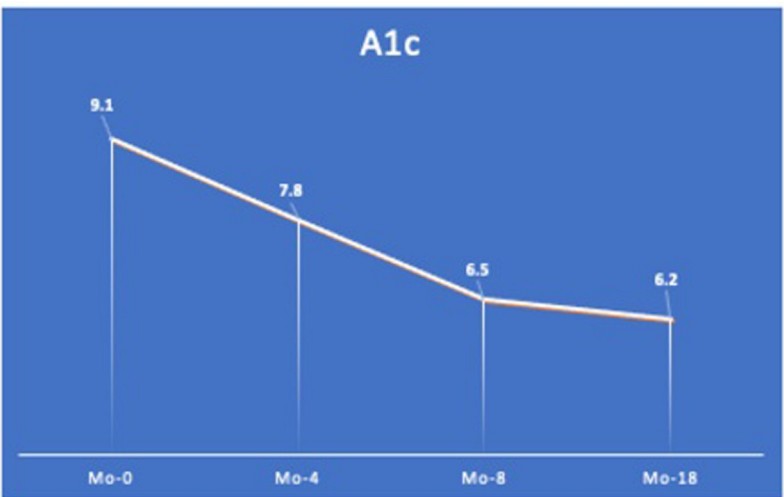

**Fig 7. Change in A1c value.** The level of A1c was reduced from 9.1 to 6.2 in 18 months (Mo) after re-establishing dental occlusion by an implant-supported fixed restoration.

associated with increased blood glucose levels [42]. Lack of or reduced chewing function could lead to less amount of fiber intake. For example, the body can absorb 130 calories out of 160 calories taken from almonds [43]. The rest of the calories (30 calories) is absorbed because the fiber in the almonds prevents early absorption in the duodenum allowing the bacteria in the jejunum and ileum to utilize the remaining 30 calories. This indicates that food with fiber, which requires a mastication function, may play an influential role in controlling calorie intake. It has also been reported that fiber reduces food intake through peptide YY3-36, which is released postprandially from the gastrointestinal L cells with GLP-1 [44]. According to Suzuki et al, masticatory function regulates postprandial blood glucose after 2 hours of eating [18] via the anorexigenic gut peptide YY and GLP-1 [16, 17]. These findings might explain how masticatory capacity involves in the regulation of blood glucose levels. The effect of masticatory performance on glucose levels, such as stimulating insulin secretion, requires further investigation.

The other mechanism of occlusal support in controlling blood glucose levels could be histaminic neurons interacting around the periodontal ligament and the masseter muscle [21]. The mesencephalic trigeminal sensory nucleus masticatory function receives proprioceptive sensory afferents of the trigeminal nerve from the masticatory muscle and periodontal ligament [21]. Indeed, these signals have been shown to stimulate the hypothalamus' satiety sensation, resulting in reduced food intake [21, 42]. It is well known that the surface area of the periodontal ligament of the posterior teeth is larger than the anterior teeth [45, 46]. Therefore, it is plausible to see a reduction in proprioceptive senses in subjects missing posterior teeth, which can reduce activation of the satiety center in the hypothalamus.

The mechanism between chewing function and masticatory muscle function is a vicious cycle that has not been clearly elucidated. Diminished or lack of masticatory capacity can result in reduced dietary protein intake, which could lead to sarcopenia [47]. Importantly, a reduction in masseter muscle thickness has been observed in patients with sarcopenia [47], and higher levels of masticatory efficiency were also negatively linked to a low level of sarcopenia [48]. The masseter muscle is one of the main mastication muscles and occlusal support directly affects its contraction, a mechanism controlled by the integrins [32]. This study clearly showed the production of integrins and masseter activity were dramatically reduced in the subjects

presenting malocclusion, crossbite, or open bite [30–32]. Therefore, lack or missing occlusal support might result in reduced masseter contractile forces on the subject. This might lead to compromised masseter activation and then reduced activation of the histamine-1 receptor in the hypothalamus, resulting in increased food intake [21]. Similarly, subjects with open bite showed a significantly shorter total duration of the chewing pattern and less masseter activity, despite molars being in contact, concerning subjects with normal dental occlusion [30]. This study supports periodontal mechanoreceptors that are more concentrated and specialized in the anterior teeth [49].

Interestingly, GLP-1 receptor agonists have been recently shown not only in reducing the level of A1c levels but also reduce the risk of stroke, all-cause mortality death, and cardiovascular disease [19]. The production of GLP-1 was increased in subjects chewing 30-time per bite [17]. Therefore, it is plausible that reduced chewing duration, such as open bite situation [30], could result in reduced insulin secretion or insufficient signal to the satiety center and/or intestinal tissues to control directly or indirectly blood glucose levels as mentioned above. Together, it is clear that oral health with proper dental occlusion plays a crucial role in maintaining general systemic health.

This study had several limitations. First, the sample size might be relatively small. However, the mean difference between the groups was statistically significant ($p = 0.0001$). Second, the subject's nutritional and/or outdoor behavior was not determined since they might play roles in the causation of metabolic disorders or controlling metabolic diseases [43, 50]. It is well-known that type of daily food intake and/or outdoor activities play important roles in the blood glucose level [51, 52]. Even if BMI at the baseline was measured, BMIs in the past could not be determined. Changing BMI values would affect blood glucose levels [53–55]. Third, the periodontal status of the subjects was not measured. It has been clearly shown that oral health conditions may affect blood glucose levels [56]. WBCs have long been associated with the periodontal health [57], finding a similar number of WBCs might indicate the subjects had a similar level of systemic inflammation that could be caused by oral infection. Finally, the number of insulin-dependent subjects was higher, but not statistically significant, in the test group than in the control group. This difference could result from the duration of diabetes in the test group. Indeed, it has been shown that islet function can be deteriorated by long-term oral hypoglycemic agents [58].

Together the individuals in each group might exhibit similar baselines in terms of contributing factors. Further studies are needed to reveal the causal relationship between occlusal support, tooth loss, and digestive function on blood glucose levels in subjects with T2D. This retrospective study was unable to reveal the causal relationship between occlusal support and blood glucose levels in T2D patients. The specific mechanism of mastication on blood glucose levels remains in additional studies.

## Conclusion

Our study showed that masticatory inefficiency due to reduced or lack of occlusal support is associated with increased A1c in patients with T2D. Our results indicated that proper mastication might play a role in controlling the blood glucose level in T2D patients. This study provides significant research information for physicians and dentists concerned with the health of diabetic patients. However, more studies are required to show a link between controlling A1c and occlusal support.

## Supporting information

**S1 Data.**
(XLSX)

## Acknowledgments

We sincerely thank Gulhan Tugrul Albayrak MD, Sisli Etfal Education and Research Hospital, for recruiting subjects, Robert Cohen D.D.S., MSD., Ph.D., University at Buffalo School of Dental Medicine Department of Periodontics and Endodontics, for editing this manuscript and Sakir Kahraman CDT (dental technicians) for the making of the prostheses.

## Author Contributions

**Data curation:** Yeter E. Bayram.

**Formal analysis:** Mehmet A. Eskan.

**Investigation:** Yeter E. Bayram, Mehmet A. Eskan.

**Methodology:** Mehmet A. Eskan.

**Validation:** Yeter E. Bayram.

**Writing – original draft:** Mehmet A. Eskan.

**Writing – review & editing:** Mehmet A. Eskan.

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
