## [Decision Letter · Decision Letter 0]

23 Feb 2023

PONE-D-22-30475Diminished or Lack of Occlusal Support is Associated with Increased Blood Glucose Levels in Patients with Type 2 DiabetesPLOS ONE

Dear Dr. ESKAN,

Thank you for submitting your manuscript to PLOS ONE. After careful consideration, we feel that it has merit but does not fully meet PLOS ONE’s publication criteria as it currently stands. Therefore, we invite you to submit a revised version of the manuscript that addresses the points raised during the review process.

We look forward to receiving your revised manuscript.

Kind regards,

Gaetano Isola, Ph.D.

Academic Editor

PLOS ONE

Journal Requirements:

2. Please ensure that you have specified (1) whether consent was informed and (2) what type you obtained (for instance, written or verbal, and if verbal, how it was documented and witnessed). If your study included minors, state whether you obtained consent from parents or guardians. If the need for consent was waived by the ethics committee, please include this information.

Additional Editor Comments (if provided):

The authors should address all comments raised by the reviewers minutely before any further assessment of the manuscript.

Reviewers' comments:

Reviewer's Responses to Questions

**Comments to the Author**

1. Is the manuscript technically sound, and do the data support the conclusions?

Reviewer #1: Partly

Reviewer #2: Partly

2. Has the statistical analysis been performed appropriately and rigorously? 

Reviewer #1: No

Reviewer #2: Yes

3. Have the authors made all data underlying the findings in their manuscript fully available?

Reviewer #1: No

Reviewer #2: Yes

4. Is the manuscript presented in an intelligible fashion and written in standard English?

Reviewer #1: No

Reviewer #2: Yes

5. Review Comments to the Author

Reviewer #1: Title with occlusal support does not specify the dental occlusal support. Henc eat would be useful to revise the title by making it more specific.

Baseline characteristics of the control and test groups should be given in a single table with columns representing values and rows showing different variables with level of statistical significance mentioned for each variable separately. Statistical significance should be mentioned for each of the variables. e.g proportion of patients taking insulin in the test and control groups, and proportion on different oral hypoglycaemic agents as they may have influenced the observed results in the two groups.

Language need major revisions with the assistance from a English language expert.

Reviewer #2: In the manuscript entitled: “Diminished or Lack of Occlusal Support is Associated with Increased Blood GlucoseLevels in Patients with Type 2 Diabetes ”, the authors examined the e the association between occlusal support and diabetic control among individuals with T2D.

The authors found that A1c levels for both groups were compared using independent sample t-tests. Differences in white blood cell counts and body mass index (BMI) were not statistically significant between groups. Statistically significant differences in glycated hemoglobin (A1c) levels were noted between the test (A1c = 9.42) and control groups (A1c = 7.48). The mean differences between the two groupswere 1.94 ± 0.39 (p = 0.0001) indicating blood glucose level (A1c) was significantly higher in participants of the test group compared to the control group (95% CI 1.159- 2.277). Moreover, blood glucose levels could be reduced (from A1c 9.1 to 6.2) after a fixed implant-supported restoration in patients with diminished occlusal support.

The authors concluded that the extent of occlusal support might have an effective role in controlling blood glucose levels among individuals with T2D.

Major comments:

In general, the idea and innovation of this study, regarding the effects of factors related to lack of occlusal support II is interesting, because the role these aspects in medicine are validated but further studies on this topic could be an innovative issue in this field could be open a creative matter of debate in literature by adding new information. Moreover, there are few reports in the literature that studied this interesting topic with this kind of study design.

The study was well conducted by the authors; However, there are some concerns to revise that are described below.

The introduction section resumes the existing knowledge regarding the important factor linked with chewing muscles and its related factors with occlusal support.

However, as the importance of the topic, the reviewer strongly recommends, before a further re-evaluation of the manuscript, to update the literature through read, discuss and must cites in the references with great attention all of those 3 recent interesting articles, that helps the authors to better introduce and discuss the role of integrin and fiber muscle activity during open bite and class II malocclusions which influences the lack of occlusal support. 1) doi: 10.3892/ijmm.2012.986. PMID: 22552408 2) doi: 10.1016/j.jelekin.2011.12.003. PMID: 22236764 3) doi: 10.2319/050615-309.1. PMID: 26502299 The authors should be better specified, at the end of the introduction section, the rational of the study and the aim of the study. In the central section, should better clarify inclusion and exclusion criteria of the selected sample.

The discussion section appears well organized with the relevant paper that support the conclusions, even if the authors should better discuss the relationship malocclusions, muscles pattern and integrin as causes of changes in chewing muscle functions and morphology. The conclusion should reinforce in light of the discussions.

In conclusion, I am sure that the authors are fine clinicians who achieve very nice results with their adopted protocol. However, this study, in my view does not in its current form satisfy a very high scientific requirement for publication in this journal and requests a revision before a futher re-evaluation of the manuscript.

Minor Comments:

Abstract:

- Better formulate the abstract section by better describing the aim of the study

Introduction:

- Please refer to major comments

Discussion

- Please add a specific sentence that clarifies the results obtained in the first part of the discussion

- Page 9 last paragraph: Please reorganize this paragraph that is not clear

6. PLOS authors have the option to publish the peer review history of their article (what does this mean?). If published, this will include your full peer review and any attached files.

Reviewer #1: No

Reviewer #2: No

---

## [Author Response · Author response to Decision Letter 0]

16 Mar 2023

The manuscript has been revised based on the journal style requirements. 

2. Please ensure that you have specified (1) whether consent was informed and (2) what type you obtained (for instance, written or verbal, and if verbal, how it was documented and witnessed). If your study included minors, state whether you obtained consent from parents or guardians. If the need for consent was waived by the ethics committee, please include this information.

It was specified under the methods section.

Comments to the Author

1. Is the manuscript technically sound, and do the data support the conclusions?

Reviewer #1: Partly

Reviewer #2: Partly

The conclusion was revised based on the data presented in the study. The revised conclusion can be easily tracked. 

2. Has the statistical analysis been performed appropriately and rigorously? 

Reviewer #1: No

Reviewer #2: Yes

More statistical analysis was performed as referee #1 mentioned “..Baseline characteristics of the control and test groups should be given in a single table with columns representing values and rows showing different variables with level of statistical significance mentioned for each variable separately..” All these new analyzed data can be seen in Table 1 in the result section. 

3. Have the authors made all data underlying the findings in their manuscript fully available?

Reviewer #1: No

Reviewer #2: Yes

Data availability was updated on the website, it can be sent to anyone without patient identification. 

4. Is the manuscript presented in an intelligible fashion and written in standard English?

Reviewer #1: No

Reviewer #2: Yes

The manuscript, as seen in yellow highlights in the text, has been dramatically changed by improving typographically and grammatically. 

5. Review Comments to the Author

Reviewer #1: Title with occlusal support does not specify the dental occlusal support. Henc eat would be useful to revise the title by making it more specific.

The title was revised accordingly. Revised title: “Mastication Inefficiency Due to Diminished or Lack of Occlusal Support is Associated with Increased Blood Glucose Levels in Patients with Type 2 Diabetes” 

It can be seen on the title page. 

Baseline characteristics of the control and test groups should be given in a single table with columns representing values and rows showing different variables with level of statistical significance mentioned for each variable separately. Statistical significance should be mentioned for each of the variables. e.g proportion of patients taking insulin in the test and control groups, and proportion on different oral hypoglycaemic agents as they may have influenced the observed results in the two groups.

All these baseline variables were included accordingly. They can be seen in Table 1 in the result section.

Language need major revisions with the assistance from a English language expert.

The manuscript language was revised accordingly. 

Reviewer #2: In the manuscript entitled: “Diminished or Lack of Occlusal Support is Associated with Increased Blood GlucoseLevels in Patients with Type 2 Diabetes ”, the authors examined the e the association between occlusal support and diabetic control among individuals with T2D.

The authors found that A1c levels for both groups were compared using independent sample t-tests. Differences in white blood cell counts and body mass index (BMI) were not statistically significant between groups. Statistically significant differences in glycated hemoglobin (A1c) levels were noted between the test (A1c = 9.42) and control groups (A1c = 7.48). The mean differences between the two groups were 1.94 ± 0.39 (p = 0.0001) indicating blood glucose level (A1c) was significantly higher in participants of the test group compared to the control group (95% CI 1.159- 2.277). Moreover, blood glucose levels could be reduced (from A1c 9.1 to 6.2) after a fixed implant-supported restoration in patients with diminished occlusal support.

The authors concluded that the extent of occlusal support might have an effective role in controlling blood glucose levels among individuals with T2D.

Major comments:

In general, the idea and innovation of this study, regarding the effects of factors related to lack of occlusal support II is interesting, because the role these aspects in medicine are validated but further studies on this topic could be an innovative issue in this field could be open a creative matter of debate in literature by adding new information. Moreover, there are few reports in the literature that studied this interesting topic with this kind of study design.

The study was well conducted by the authors; However, there are some concerns to revise that are described below.

The introduction section resumes the existing knowledge regarding the important factor linked with chewing muscles and its related factors with occlusal support.

However, as the importance of the topic, the reviewer strongly recommends, before a further re-evaluation of the manuscript, to update the literature through read, discuss and must cites in the references with great attention all of those 3 recent interesting articles, that helps the authors to better introduce and discuss the role of integrin and fiber muscle activity during open bite and class II malocclusions which influences the lack of occlusal support. 1) doi: 10.3892/ijmm.2012.986. PMID: 22552408 2) doi: 10.1016/j.jelekin.2011.12.003. PMID: 22236764 3) doi: 10.2319/050615-309.1. PMID: 26502299

All these articles were included accordingly in the introduction (page 5, lines 91-98) and discussion section (page 12, lines 250-263) 

1) doi: 10.3892/ijmm.2012.986. PMID: 22552408 (ref# 32)

2) doi: 10.1016/j.jelekin.2011.12.003. PMID: 22236764 (ref#30)

3) doi: 10.2319/050615-309.1. PMID: 26502299 (Ref# 31)

The authors should be better specified, at the end of the introduction section, the rational of the study and the aim of the study. In the central section, should better clarify inclusion and exclusion criteria of the selected sample.

The study rationale has been revised at the end of the introduction.

Inclusion and exclusion criteria were revised accordingly. 

The discussion section appears well organized with the relevant paper that support the conclusions, even if the authors should better discuss the relationship malocclusions, muscles pattern and integrin as causes of changes in chewing muscle functions and morphology. The conclusion should reinforce in light of the discussions.

The relationship between malocclusion and muscle was discussed accordingly. It can be seen on page 12, lines 250-263. 

In conclusion, I am sure that the authors are fine clinicians who achieve very nice results with their adopted protocol. However, this study, in my view does not in its current form satisfy a very high scientific requirement for publication in this journal and requests a revision before a futher re-evaluation of the manuscript.

Minor Comments:

Abstract:

- Better formulate the abstract section by better describing the aim of the study

 The abstract was revised accordingly. 

Introduction:

- Please refer to major comments

It was revised accordingly

Discussion

- Please add a specific sentence that clarifies the results obtained in the first part of the discussion

The first part of the discussion has been revised accordingly. All the changes that were made in the discussion can be tracked in the highlighted version of the paper. 

- Page 9 last paragraph: Please reorganize this paragraph that is not clear

It was corrected and it can be traced on page 11 lines 235-240

6. PLOS authors have the option to publish the peer review history of their article (what does this mean?). If published, this will include your full peer review and any attached files.

Do you want your identity to be public for this peer review? For information about this choice, including consent withdrawal, please see our Privacy Policy.

Reviewer #1: No

Reviewer #2: No

It should be okay for our identity to be public. 

Kind regards,

Mehmet A Eskan DDS PhD

Clin Asst Prof

University at Buffalo School of Dental Medicine

---

## [Decision Letter · Decision Letter 1]

28 Mar 2023

Mastication Inefficiency Due to Diminished or Lack of Occlusal Support is Associated with Increased Blood Glucose Levels in Patients with Type 2 Diabetes

PONE-D-22-30475R1

Dear Dr. ESKAN,

We’re pleased to inform you that your manuscript has been judged scientifically suitable for publication and will be formally accepted for publication once it meets all outstanding technical requirements.

Kind regards,

Gaetano Isola, Ph.D.

Academic Editor

PLOS ONE

Additional Editor Comments (optional):

The authors have well addressed all concerns raised by both reviewers. No further issues are needed.

Reviewers' comments:

Reviewer's Responses to Questions

**Comments to the Author**

1. If the authors have adequately addressed your comments raised in a previous round of review and you feel that this manuscript is now acceptable for publication, you may indicate that here to bypass the “Comments to the Author” section, enter your conflict of interest statement in the “Confidential to Editor” section, and submit your "Accept" recommendation.

Reviewer #2: All comments have been addressed

2. Is the manuscript technically sound, and do the data support the conclusions?

Reviewer #2: Yes

3. Has the statistical analysis been performed appropriately and rigorously? 

Reviewer #2: Yes

4. Have the authors made all data underlying the findings in their manuscript fully available?

Reviewer #2: Yes

5. Is the manuscript presented in an intelligible fashion and written in standard English?

Reviewer #2: Yes

6. Review Comments to the Author

Reviewer #2: All of the comments were well analyzed and solved. No further issues are needed regarding all issues

7. PLOS authors have the option to publish the peer review history of their article (what does this mean?). If published, this will include your full peer review and any attached files.

Reviewer #2: No
